# ADAM17—A Potential Blood-Based Biomarker for Detection of Early-Stage Ovarian Cancer

**DOI:** 10.3390/cancers13215563

**Published:** 2021-11-06

**Authors:** Christoph Rogmans, Jan Dominik Kuhlmann, Gerrit Hugendieck, Theresa Link, Norbert Arnold, Jörg Paul Weimer, Inken Flörkemeier, Anna-Christina Rambow, Wolfgang Lieb, Nicolai Maass, Dirk O. Bauerschlag, Nina Hedemann

**Affiliations:** 1Department of Gynecology and Obstetrics, Christian-Albrechts-University Kiel and University Medical Center Schleswig-Holstein Campus Kiel, 24105 Kiel, Germany; christoph.rogmans@uksh.de (C.R.); gerrithugendieck@hotmail.de (G.H.); Norbert.Arnold@uksh.de (N.A.); Joerg-Paul.Weimer@uksh.de (J.P.W.); inken.floerkemeier@uksh.de (I.F.); anna-christina.rambow@uksh.de (A.-C.R.); nicolai.maass@uksh.de (N.M.); dirk.bauerschlag@uksh.de (D.O.B.); 2Department of Gynecology and Obstetrics, Medical Faculty and University Hospital Carl Gustav Carus, Technische Universität Dresden, 01219 Dresden, Germany; Jan.Kuhlmann@uniklinikum-dresden.de (J.D.K.); theresa.link@uniklinikum-dresden.de (T.L.); 3National Center for Tumor Diseases (NCT), Dresden, Germany: German Cancer Research Center (DKFZ), Heidelberg, Germany; Faculty of Medicine and University Hospital Carl Gustav Carus, Technische, Universität Dresden, Dresden, Germany, Helmholtz-Zentrum Dresden-Rossendorf (HZDR), Dresden, Germany; 4German Cancer Consortium (DKTK), Dresden and German Cancer Research Center (DKFZ), 69120 Heidelberg, Germany; 5Institute of Epidemiology, Christian-Albrechts-University Kiel and University Medical Center Schleswig-Holstein Campus Kiel, 24105 Kiel, Germany; wolfgang.lieb@epi.uni-kiel.de

**Keywords:** ovarian cancer, ADAM17, serum, ascites, tumor marker, early detection

## Abstract

**Simple Summary:**

Ovarian cancer has the highest lethality among gynecological tumors. Therefore, it is essential to find reliable biomarkers to improve early detection. This is the first report describing ADAM17 detection in serum and ascites fluid of ovarian cancer patients. A high ADAM17 concentration in serum at primary diagnosis is associated with early FIGO stages and predicts complete resection of the tumor mass. In addition, ADAM17 and CA-125 complement each other, especially in the diagnosis of early stages. In summary, ADAM17 appears to be a promising screening marker for detecting early-stage ovarian cancer.

**Abstract:**

Ovarian cancer has the highest mortality rate among gynecological tumors. This is based on late diagnosis and the lack of early symptoms. To improve early detection, it is essential to find reliable biomarkers. The metalloprotease ADAM17 could be a potential marker, as it is highly expressed in many solid tumors, including ovarian and breast cancer. The aim of this work is to evaluate the relevance of ADAM17 as a potential diagnostic blood-based biomarker in ovarian cancer. Ovarian cancer cell lines IGROV-1 and A2780, as well as primary patient-derived tumor cells obtained from tumor tissue and ascitic fluid, were cultured to analyze ADAM17 abundance in the culture supernatant. In a translational approach, a cohort of 117 well-characterized ovarian cancer patients was assembled and ADAM17 levels in serum and corresponding ascitic fluid were determined at primary diagnosis. ADAM17 was quantified by enzyme-linked immunosorbent assay (ELISA). In the present study, ADAM17 was detected in the culture supernatant of ovarian cancer cell lines and primary cells. In addition, ADAM17 was found in serum and ascites of ovarian cancer patients. ADAM17 level was significantly increased in ovarian cancer patients compared to an age-matched control group (*p* < 0.0001). Importantly early FIGO I/II stages, which would not have been detected by CA-125, were associated with higher ADAM17 concentrations (*p* = 0.007). This is the first study proposing ADAM17 as a serum tumor marker in the setting of a gynecological tumor disease. Usage of ADAM17 in combination with CA-125 and other markers could help detect early stages of ovarian cancer.

## 1. Introduction

With more than 100,000 deaths per year worldwide, ovarian cancer is the most lethal cancer amongst all gynecological malignancies [1]. The vast majority of ovarian cancers are diagnosed in advanced stages due to unspecific symptoms and the lack of reliable biomarkers [2]. The second challenge in ovarian cancer is the resistance to chemotherapeutic treatment, which often leads to recurrent disease [3]. So far, no screening for ovarian cancer has been established.

Treatment of advanced ovarian cancer is challenging because tumor-free resection (R0-surgery) is less likely, as the tumor has already spread widely in the peritoneal cavity. Because tumor-free resection, ever since, has led to better survival rates, it is of major clinical importance to find novel biomarkers indicating malignant changes of the ovaries in early stages [4]. In recent years, attempts to find a valid screening marker have failed. The low prevalence of ovarian cancer requires high specificity to achieve a clinically acceptable positive predictive value of at least 10%. Moreover, most tumor markers in the early stages show low sensitivity (stage I 40–50%) [5]. Conversely, benign diseases such as endometriosis can also lead to increased levels of cancer antigen 125 (CA-125), which significantly limits specificity [6].

Although CA-125 is an accepted serological tumor marker for ovarian cancer, its application for screening approaches is controversial due to the high percentage of false-positive results. Thus, it is mainly used for therapy monitoring, during chemotherapeutic treatment. High CA-125 levels do not necessarily indicate a specific tumor entity or the infestation of a particular organ. Increased CA-125 values are also found in other malignant diseases such as endometrial carcinoma, tumors of the intestinal tract, and certain pancreatic tumors [7]. Furthermore, an increase of this marker is not exclusively specific for a malignant event. Elevated CA-125 levels can be detected in 1% of all healthy women [8]. Although the highest serum levels of CA-125 are found in patients with ovarian cancer, an increase in serum CA-125 may be associated with benign lesions or under certain physiological conditions, including pregnancy, endometriosis, or menstruation [6].

Continuous screening of CA-125 did not improve patient outcomes in terms of survival but reduced mental health issues [9]. Thus, the identification of novel markers to detect ovarian cancer is highly warranted.

A new approach is the detection of the tumor marker HE4 (human epididymis protein 4). HE4 has a similar sensitivity to CA-125 [10]. However, it differs in its increased specificity in regard to the distinction from malignant vs. benign diseases [11]. Both parameters are combined in the Risk of Ovarian Malignancy Algorithm (ROMA index) [12]. This index categorizes patients into a high- or low-risk group. Despite this improvement, the ROMA index often fails to detect a low tumor burden, rendering an early diagnosis of ovarian cancer a clinical challenge.

Physiologically, a disintegrin and metalloprotease 17 (ADAM17) is a membrane-bound metalloproteinase that plays an important role in embryogenesis and tissue regeneration, mainly by shedding growth factors and concomitant activation of the epidermal growth factor receptor (EGFR) signaling pathways [13]. An enhanced processing of these factors and high expression of ADAM17 in tumors has been shown in a variety of solid tumors including ovarian cancer [14]. We have recently shown that the ADAM17 protein is expressed by ascites derived from ovarian cancer cells [15]. Many of the ADAM17 substrates, such as Nectin-4 and HB-EGF (heparin-binding epidermal growth factor-like factor) have already been investigated as potential tumor markers for ovarian cancer [16]. However, ADAM17 is the protease that cleaves and releases a large number of these factors [17]. In breast cancer patients, ADAM17 expression in tissue correlates with an increased risk for metastasis and poor survival [18].

The detection of ADAM17 in serum has so far only been described for inflammatory diseases such as rheumatoid arthritis or endotoxic shock syndrome in sepsis, but not in connection with a malignant gynecological disease such as ovarian cancer [19,20]. Only Walkiewicz et al. were able to detect ADAM17 in the blood of colorectal cancer patients [21]. For a variety of solid tumors, including ovarian cancer, elevated levels of ADAM17 have been detected by immunohistochemistry, flow cytometry, and real-time PCR, respectively. However, it has not yet been detected in ovarian cancer patients’ sera [22,23,24].

Because of its high expression in tumors even in the early stages, the evaluation of ADAM17 as a potential blood-based screening marker for early detection of ovarian cancer seemed rational for us and will be examined more closely [24] to potentially improve detection of ovarian cancer in earlier stages.

The objective of this study was to investigate whether:(i)ADAM17 is released by ovarian cancer cell lines and patient-derived cells into the culture supernatant(ii)ADAM17 is differentially detectable in serum samples of ovarian cancer patients compared to healthy controls(iii)ADAM17 has clinical relevance e.g., in terms of blood-based detection of early-stage ovarian cancer

For the first time, the present study identified the metalloprotease ADAM17 in the serum of ovarian cancer patients and indicates a possible application as a marker for early-stage disease.

## 2. Materials and Methods

### 2.1. Ethics Statement

This research was approved by the Institutional Review Board of the University Medical Center Schleswig–Holstein, Campus Kiel (AZ: B327/10) according to the Declaration of Helsinki. Written informed consent was obtained from all patients.

### 2.2. Cell Culture

The ovarian cancer cell line A2780 was purchased from Sigma-Aldrich and the human ovarian adenocarcinoma cell line IGROV-1 was obtained from American Type Culture Collection (ATCC). The ovarian cancer cell lines were cultured in RPMI-1640 medium (Biochrom GmbH, Berlin, Germany) #FG 1415), supplemented with including 50 L-glutamine (Sigma-Aldrich, St. Louis, MO, USA) with 10% fetal bovine serum and penicillin-streptomycin (60 IU (µg/mL)). Cell lines were authenticated by short tandem repeat (STR) DNA profile analysis [25]. The profile analysis was performed before and during cultivation. Routinely, mycoplasma contamination was ruled out using the MycoAlert kit™ (Lonza, Basel, Switzerland # LT07).

To investigate the ADAM17 concentration in culture supernatants of ovarian cancer cells 1.5 × 10^6^ cells were seeded in 6-well plates and incubated at 37 °C and 5% CO_2_. After 72 h the culture supernatant was removed and examined for its ADAM17 concentration by enzyme-linked immunosorbent assay (ELISA). To determine ADAM17 in OvCa cells, Igrov-1 cells were cultured for 72 h and lysates were generated as described previously [26].

### 2.3. Primary Cells and Ascites Fluids

Primary cells of ovarian cancer patients with advanced tumor stage were collected during surgery. Cells were isolated from the tumor tissue as well as from the ascites fluid. Ascites fluid was centrifuged (348× *g*, 10 min) and the pellet dissolved in 12 mL RPMI-1640 medium [27]. The cells were seeded in a tissue culture flask, expanded, and used for experiments as soon as a confluence of 80% was reached. Primary tumor cells were extracted from the tumor tissue and transferred to the supplemented RPMI medium described above. The polyploid character of the primary cells was checked by fluorescence in situ hybridization (FISH) using the three-color probe TERC (3q26)/MYC (8q24)/SE 7 TC (Kreatech/Leica, Wetzlar, Germany # KBI-10704). A signal pattern that deviates from the inconspicuous diploid pattern was confirmed in 60–100% of the examined patient-derived cells. Since ovarian carcinoma cells have various rearrangements of the chromosomes and thus deviate from the inconspicuous, diploid character, we evaluate the detection of abnormal signal patterns as a tumor cell.

### 2.4. ADAM17 ELISA

To detect ADAM17 in culture supernatants and serum, the human ADAM17 Duoset Sandwich ELISA (R&D Systems, Minneapolis, MN, USA #DY930) was used according to the modified test instructions:

The capture antibody was diluted to a concentration of 2 µg/mL and the corresponding amount was applied to the 96-well NUNC-IMMUNO plate and incubated overnight. The next day, the plate was washed with wash buffer and blocked with Reagent Diluent. The plate was incubated for one hour at room temperature and rinsed again with wash buffer. Subsequently, standards and samples were applied as duplicates and incubated for a further two hours. After further rinsing with wash buffer, the detection antibody was diluted to a concentration of 0.5 µg/mL and applied to the plate. After two hours incubation, the unbound detection antibody was washed off and the plate was treated with streptavidin HRP diluted in Reagent Diluent. Again, after 20 min incubation at room temperature and protection from light, the plate was rinsed with wash buffer. The substrate solution was added and after 15 min the reaction was stopped with the stop solution. The optical density (OD) was measured at 450 nm with a microtiter plate reader (Infinite 200, Tecan, Männedorf, Switzerland). To quantify ADAM17 substrate levels of Nectin-4 and HB-EGF the following ELISA kits were used according to the manufacturer’s instructions: Human Nectin-4 DuoSet ELISA (R&D Systems, #DY2659) and Human HB-EGF DuoSet ELISA (R&D Systems, #DY259B). The test was performed according to the instructions.

### 2.5. Cell Lyses, Westernblot and Densitometry

In brief, total cell lysates as well as cell-free ascites samples were denatured using either ROTI^®^Load2 (4×) buffer (# K930.1) and heated for 5 min at 95 °C. Samples were loaded on a 4–12% gradient Bis-Tris gel (NuPAGE, Invitrogen by ThermoFisher Scientific, Waltham, MA, USA # NP0336) and separated by electrophoresis (90 min, 200 V at 4 °C). After being transferred to Polyvinylidendifluorid (PVDF) membranes (Merck Millipore, Burlington, MA, USA, Immobilon-Fl, # IPFL 00010), membranes were blocked using 5% of milk powder (Carl Roth, Karlsruhe, Germany) in TBS-T for 1 h. Following overnight incubation at 4 °C using (anti-ADAM17 rabbit-Ab 1:2000 (Abcam, Cambridge, United Kingdom #ab39162) primary antibody, horseradish peroxidase-conjugated secondary antibodies (CellSignaling, Danvers, MA, USA #7074) were incubated at room temperature for 1 h. For detection, the ECL Substrate Kit was used (BioRAD Claritywestern ECL Substrate, Hercules, CA, USA #170-5061). Semiquantitative densitometry was performed using the QuantityOne software (Biorad). For comparison of relative band intensities, the strongest band intensity was set to 1.00.

### 2.6. Patient Characteristics und Serum Samples

A standard operating procedure (SOP) was developed as a guideline for the collection, aliquoting, and storage of blood samples. Blood samples were collected from the Department of Gynecology at the University Hospital Schleswig–Holstein, Kiel Campus, as part of routine blood sampling. The samples were processed within 2 h after blood sampling. Centrifugation was performed at 3000 rpm for 10 min. Samples were stored in an 80 °C freezer and managed by the laboratory’s internal database. The evaluated parameters included: age at diagnosis, survival-rate, Fédération Internationale de Gynécologie et d’Obstétrique (FIGO) stage, histology of the tumor tissue, tumor grading, resection of the tumor (R0 vs. R1) by surgery, progression-free survival (Table 1).

Controls were derived from a community-based sample from the Kiel area and were age-matched to the patient group. For the control cohort, a detailed medical history was also gathered and only controls who were negative for ADAM17-associated diseases (tumor diseases such as breast or ovarian cancer, inflammatory diseases such as chronic inflammatory bowel disease such as Crohn’s disease and colitis ulcerosa, or autoimmune diseases such as rheumatoid arthritis and psoriasis) were included in the present analyses. The control samples were provided by the PopGen 2.0 Network at the UKSH in Kiel [28].

### 2.7. Statistics

All patient data collected were encrypted, anonymized, and evaluated with the statistical programs SPSS and Graphpad Prism. (Version 8) (La Jolla, CA, USA). Cell culture data were summarized using Microsoft Excel (Version 2019) (Redmond, WA, USA) and evaluated in Graphpad Prism. A normal distribution test was performed using Shapiro–Wilk and Kolmogorov–Smirnov tests. As serum samples were not normally distributed, the Mann–Whitney *U*-test was used for two independent samples and the Kruskal–Wallis test for three and more independent samples. To evaluate the significance of ADAM17 as a tumor marker, ROC curves were generated and the Youden index was performed to determine suitable cut-off values.

## 3. Results

### 3.1. ADAM17 Is Detectable in Culture Supernatants of Ovarian Cancer Cell Lines and Patient Derived Tumor Cells

ADAM17 is physiologically located at the cell membrane, in vesicles or in the cytoplasm, therefore we firstly investigated, if ADAM17 can be detected in cell culture supernatants as a result of active or passive release from ovarian cancer cells [29]. Scheff et al. showed that different levels of ADAM17 are detectable in the cell supernatant of oral squamous cell carcinoma even in the dissolved form [30]. Therefore, after 72 h cultivation, cell culture supernatant of 80% confluent cell layers was collected. This conditioned medium (CM) was analyzed using ELISA.

We analyzed two ovarian cancer cell lines: IGROV-1 and A2780. Moreover, we extracted cells from tumors of two ovarian cancer patients (UF-169-tumor, UF-192-tumor) and one corresponding ascites fluid (UF-169-Asc.), derived at primary diagnosis. In all investigated supernatants, we detected ADAM17 levels between 1500 pg/mL and 3500 pg/mL and thus confirmed that ADAM17 is detectable in conditioned medium of ovarian cancer cell lines and patient-derived cells (Figure 1). This observation was the prerequisite to studying ADAM17 concentrations in body fluids such as serum and ascites.

### 3.2. ADAM17 Is Detectable in Serum and Corresponding Ascites of Ovarian Cancer Patients

Since we confirmed the release of ADAM17 in the supernatant, we hypothesized that ADAM17 could also be detected in serum and cell-free ascites of ovarian cancer patients. For this approach at primary diagnosis drawn serum and ascites fluid from 10 ovarian cancer patients were analyzed by ADAM17 sandwich ELISA. We found different amounts of ADAM17 in serum vs. ascites of the individual patients (Figure 2). Interestingly, the magnitude of ADAM17 levels of corresponding ascites and serum samples followed a similar trend; i.e., Patient 5 (P-OC-5) revealed the highest amounts of ADAM17 in both serum (3859 pg/mL) and ascites (4246 pg/mL), whereas patient 2 (P-OC-2) had 85 pg/mL in serum and 177 pg/mL in ascites. Moreover, ADAM17 levels in ascites samples were on average 2.5 times higher than ADAM17 amounts in serum (*p* = 0.0049). For validation, we performed Western blot analyses using the identical ascites samples (Appendix A; band description: Appendix A; uncropped western blots: Appendix A). ADAM17 was predominantly observed at a molecular weight of ~55–60 kDa. This fragment was previously described as a secondary cleavage product of ADAM17 by other proteases such as matrix metalloproteases (MMPs) [31,32]. As the presence and activity of MMPs in ascites fluid was shown by several studies, this form was further analyzed and quantified by densitometry to be comparable to ELISA results [33,34]. We obtained identical tendencies of ADAM17 levels using this complementary method; except for sample 10 (strongly hemorrhagic).

Taken together, ADAM17 serum levels differ inter-individually with overall higher ADAM17 concentrations in ascites than in serum, respectively.

### 3.3. ADAM17 a Novel Marker for Early Detection of Ovarian Cancer?

To investigate if ADAM17 could function as a novel diagnostic marker for ovarian cancer at primary diagnosis, we collected and characterized serum of 117 ovarian cancer patients. The samples were tested for their ADAM17 concentration by sandwich ELISA (Figure 3A). As a control cohort, 100 age-matched female donors with no history of malignancies were probed. Direct comparison of these two cohorts shows a significant difference of ADAM17 levels between the patient cohort vs. control cohort (*U*-test, *p* < 0.001); ADAM17 values are significantly higher in the patient group than in the control group (average: patients diagnosed ovarian cancer: 3753 pg/mL, control: 2197 pg/mL). To determine suitable ADAM17 cut-off values, we performed ROC curve analyses (Figure 3B) and generated the Youden index. At an ADAM17 value of 39.792 pg/mL, the calculated cut-off showed a sensitivity of 96.58% and specificity of 60.91% with an AUC of 0.78 (CI = [0.72–0.85]). These findings underline that ADAM17 is a sensitive marker to discriminate diseased from non-diseased patients.

Since we measured elevated ADAM17 levels in patients suffering from ovarian cancer, we proceeded with investigating whether ADAM17 levels correlate with other clinical parameters, such as the FIGO stage, histological subtype, or the postoperative resection status.

#### 3.3.1. Early-Stage Patients Have Higher ADAM17 Serum Levels

The clinical stages of ovarian cancer, classified according to the FIGO classification, are decisive for diagnosis. Therefore, the concentration of ADAM17 was correlated with individual FIGO stages. Since the cohort is very heterogeneous considering all histological subtypes, the serous subtype with the largest number of cases was selected for all further statistical investigations. For this purpose, blood-based ADAM17 levels in the prognostically more favorable early stages I and II (12 cases) were combined and compared to the advanced stages III and IV (72 cases). Strikingly there was a significant difference between FIGO I/II and FIGO III/IV regarding the detectable ADAM17 amounts (*U*-test, *p* = 0.007) (Figure 4A). After applying ROC analysis, a cut-off value of 2588.5 pg/mL was calculated. This corresponds to a sensitivity of 72.22%; and a specificity of 75.00% with an AUC of 0.74 (CI = [0.60–0.88]) for the blood-based discrimination between FIGOI/II vs. FIGOIII/IV ovarian cancer patients (Figure 4B).

Interestingly, the comparison between the early stages and the control cohort is also statistically significant (*U*-test, *p* < 0.0001) (Figure 4C). To determine suitable ADAM17 cut-off values, we performed ROC curve analyses (Figure 4D) and generated the Youden index. At an ADAM17 value of 569.8 pg/mL the calculated cut-off showed a sensitivity of 100% and specificity of 72.7% with an AUC 0.84 (CI = [0.77–0.91]).

The fact, that the early stages show increased ADAM17 concentrations, renders the possibility of an earlier diagnosis.

For a functional validation of ADAM17 activity in these samples, we additionally investigated if the ADAM17 substrates Nectin-4 and HB-EGF are detectable (Appendix A). Therefore, we tested five sera of early stages (FIGO I/II) against five sera of late-stage OvCa patients (FIGO III/IV) and quantified their substrate levels as a surrogate for ADAM17 activity. Both substrates demonstrated higher levels in early compared to late FIGO stages. Notably, the differences of Nectin-4, a substrate, which is also proteolytically cleaved by ADAM10 were less pronounced (mean: 342.7 vs. 92.50) compared to HB-EGF (mean: 14.26 vs. 2.974) being the sole substrate of ADAM17 (Mann–Whitney test *p* = 0.0278) [13,17,35] (Appendix A). Thus, not only ADAM17 serum levels were increased in early FIGO stages but also ADAM17 activity was strongly upregulated in these patients.

#### 3.3.2. ADAM17 Is Highly Expressed in the Endometrioid Subtype

To answer the question, if ADAM17 levels correlate with histological subtypes of ovarian cancer, we compared ADAM17 levels of 117 patients including 84 serous, 18 endometrioid, 6 clear cell, and 9 other histological subtypes (Figure 5B). The median ADAM17 concentration of endometrioid subtype was 6008 pg/mL notedly higher compared to other histologically subtypes especially the serous subtype: 3281 pg/mL (*p* = 0.082) (Figure 5A).

#### 3.3.3. Higher ADAM17 Level Predicts Optimal Primary Tumor Debulking

One of the most important prognostic parameters in ovarian cancer is the complete surgical removal of the tumor mass. In order to investigate whether ADAM17 may predict primary debulking efficiency, the proportion of patients without macroscopic tumor remnants (48 cases) was compared to those with residual tumor burden (35 cases), with regard to ADAM17 serum levels. Remarkably, we found a higher concentration in those patients in which complete tumor debulking could be achieved, reflecting the situation on behalf of the higher concentration in FIGO I/II where a complete debulking was more likely (*U*-test, *p* = 0.048).

### 3.4. Comparison between ADAM17 and CA-125

Preoperatively, no correlation was evident between ADAM17 and CA-125 at primary diagnosis (correlation: r = 0.061, *p* = 0.599). Therefore, it can be assumed that both are independent markers. In our cohort, 107/117 patients had both parameters assessed. Of these, 16 patients had relatively low CA-125 levels (<100 U/mL, 15th percentile of CA-125 levels) [36] (Figure 6). Strikingly, the majority of patients (14/16) had significantly elevated ADAM17 levels (>862 pg/mL, 75th percentile of control group). Seven of these patients were in an early stage of the disease.

## 4. Discussion

In most cases, ovarian cancer is diagnosed at an advanced stage of disease and is associated with a poor prognosis. The cure rate is highest for FIGO stages I and II. Thus, early detection would significantly improve long-term survival and increase the chances of cure [37].

Therefore, the establishment of a diagnostic tumor marker for ovarian cancer, which indicates a malignant event especially in the early stages, is crucial due to the absence of early symptoms in this disease.

Accordingly, we focused our research on the metalloprotease ADAM17, playing an important role in the initiation and progression of malignancies. Increased expression levels of the ADAM17 protein have been described in many solid tumors, including lung, gastric, renal, colon, pancreatic, and ovarian cancer [38,39]. ADAM17 cleaves a considerable number of substrates proteolytically and thus plays an important role in the physiological tissue development of organs and tissue regeneration [40,41]. Overexpression or increased activation of ADAM17 in tumor cells has been associated with the initiation and progression of carcinomas, especially when EGFR is activated [40]. In comparison, “high risk tumors” showed significantly higher ADAM17 expressions than “low risk tumors” [18]. High ADAM17 expression was also associated with increased lymph node involvement and increased tumor mass in breast cancer [42]. Multiple studies investigated ADAM17 tissue expression in ovarian cancer. ADAM17 protein levels, analyzed by immunohistochemistry, were strongly increased in patients compared to healthy ovarian tissue [24,43,44]. As for these convincing results, we focused our research on liquid samples such as culture supernatants, ascites, and serum, to investigate if ADAM17 could be likewise detected in these liquids. This is of particular interest as serum could be obtained minimal-invasively before surgery and is indicative of the diagnosis.

In this study, the cell culture experiments confirmed that ADAM17 is detectable in the supernatant of ovarian cancer cell lines and patient-derived cells after culturing. Scharfenberg et al. recently demonstrated that ADAM17 is shed by ADAM8 and can be released proteolytically from the cell surface in cardiac muscle cell lines [45]. Mongaret et al. were also able to identify for other representatives of the ADAM family, such as ADAM9, that proteolytic cleavage occurs at the cell surface and secretion into the cell supernatant takes place [46]. These findings support our results that ADAM17 is released by ovarian cancer cells.

Various studies have already shown that ADAM17 plays a central role in several diseases and can be detected in different body fluids. Elevated ADAM17 levels have been shown in the serum of Alzheimer’s disease, rheumatoid arthritis, and ANCA-associated vasculitis, in cerebrospinal fluids of patients with neoplastic meningitis, in kidney cyst fluids of polycystic kidney disease, in ascites in sepsis with extensive peritonitis, and even in the context of colorectal cancer [19,20,21,47,48]. In this study, we achieved detection of ADAM17 in serum as well as in ascites fluid of ovarian cancer patients. This is the first time detecting elevated ADAM17 levels in the context of malignant gynecological disease in serum.

The observation of different concentrations of ADAM17 in serum and ascites of the individual patients at primary diagnosis could indicate an exchange between serum and peritoneal fluid or similar processes in both compartments. As assumed, ADAM17 levels in ascites are significantly higher than in the serum of the corresponding patients, confirming the trends of individual patients are quite comparable. Similar results were obtained by Kermarrec et al. who also measured higher ADAM17 levels in ascites fluid compared to serum in sepsis patients [20]. Poersch et al. showed that a wide range of proteins and cytokines are secreted into the ascites fluid in the case of serous ovarian cancer [49]. Measurements of cytokines in serum and ascites of ovarian cancer patients showed similar results. L1 cell adhesion molecule (L1-CAM) and Nectin-4 were also detected in higher amounts in the ascites fluid than in serum [17,50]. In the case of advanced tumor stage, this indicates that tumor-derived ADAM17 could primarily be released into malignant ascites and then secondarily transferred from ascites into the patient’s blood. In our study, we confirmed the presence of Nectin-4 and HB-EGF in patient serum and even demonstrate a correlation of substrates with ADAM17 levels, suggesting that ADAM17 is also active in the corresponding patients.

These initial results suggested that ADAM17 may be a suitable marker for the detection of ovarian cancer. Therefore, we tried to detect ADAM17 in serum samples of ovarian cancer patients. After an initial evaluation of all histological subtypes, the cohort was restricted to high-grade serous cystadenocarcinomas. Nevertheless, investigation of ADAM17 as a tumor marker in a larger subgroup of endometrioid ovarian cancer would be of interest.

Subsequently, attempts were made to validate the applicability of ADAM17 as a tumor marker in the screening setting. Initially, we observed that ADAM17 has a good diagnostic quality regarding the discrimination between the patient cohort and the control group. At a cut-off value of 39.792 pg/mL, sensitivity was 96.58% and specificity was 60.91%. Although this demonstrates that ADAM17 is a very sensitive marker, the positive predictive value is too small to consider screening with ADAM17 alone.

There was a clear correlation between ADAM17 and clinical parameters such as FIGO stage, and residual tumor burden after primary debulking.

Tumor spread is an important clinical prognostic factor and has profound consequences for postoperative therapy and clinical outcome [51]. Relevant tumor markers such as CA-125 and CA 72–4 tend to be elevated in advanced FIGO stages compared to the early stages [52]. One explanation for this is that the level of CA-125 is associated with greater changes in tumor mass and thus changes congruently with the FIGO stage [53,54]. In this study, a significant difference between FIGO I/II and FIGO III/IV could be shown regarding the detectable ADAM17 levels (*p* = 0.007). ADAM17 concentrations were 50% higher in the early stages compared to the advanced stages, reacting conversely to CA-125. Therefore, we hypothesize that increased ADAM17 released into the bloodstream is rather a surrogate for tumor progression in early-stage disease than a surrogate for increased tumor burden of advanced disease, pointing to the complex regulation of ADAM17. To investigate if ADAM17 is also functionally active in these patients, we additionally analyzed the ADAM17 substrates Nection-4 and HB-EGF as a surrogate for ADAM17 activity. Importantly, we found higher levels of both substrates in early-stage patients compared to late-stage patients. This effect was even more pronounced in HB-EGF levels compared to Nectin-4, as HB-EGF is solely processed by ADAM17 [39]. Thus, HB-EGF levels can be interpreted as a surrogate marker for ADAM17 activity. Because of these findings, ADAM17 could be a new marker for an earlier diagnosis, as the early stages are largely asymptomatic and difficult to identify. However, the low specificity of ADAM17 should also be mentioned here, which makes it essential to also consider clinical findings. Furthermore, the application of ADAM17 as a clinical parameter for the detection of ovarian cancer must be qualified by the fact that other diseases, such as Alzheimer’s disease, rheumatoid arthritis, and ANCA-associated vasculitis, can also lead to elevated ADAM17 levels. Therefore, a combination of ADAM17 and CA-125, especially in the identification of early stages would be conceivable.

The postoperative residual tumor has been known as a prognostic factor since the 1970s [55]. Bristow et al. were able to show that the size of the postoperative residual tumor is inversely related to overall survival [4]. In the late 1990s, Gonzalez et al. concluded that the preoperative levels of CA-125 in patients with residual tumor after tumor reduction surgery were significantly higher than in patients who had undergone complete resection [56]. This is controversial, as Rossi and colleagues were unable to establish a correlation between preoperatively determined CA-125 levels and postoperative residual tumor levels [57]. In the present study, statistical analysis showed that the ADAM17 concentration was significantly lower when a macroscopic tumor residue remains (*p* = 0.048). It is interesting that ADAM17 levels in patients with no residual disease, as well in patients with early stages, are significantly higher than in the reference group. This can be explained by a close correlation of the parameters since tumor-free resection is more likely to be achieved in the early stages than in advanced tumor stages. A review of the patient cohort showed that, as expected, about 33% of the macroscopically R0-resected patients had FIGO stage I/II, and 95% of the macroscopically R1-resected patients had FIGO stage III/IV.

It is known that the tumor stroma is an active factor in carcinogenesis and contains a variety of cell types, such as vascular cells, cancer-associated fibroblasts, and inflammatory cells, which play a key role in invasion and migration [58]. By creating an environment that promotes tumor growth and suppresses the immune system, an autocrine paracrine communication loop is established. This causes a mutual amplification of growth and migration signals [59,60]. Essential elements of these signaling pathways are matrix metalloproteases, to which ADAM17 is a major player. Their expression can be detected in many repair or remodeling processes and in diseased or inflamed tissue [61,62]. Moreover, for ADAM17 substrates such as Nectin-4 and HB-EGF, it was shown that increased levels contribute to tumor proliferation [63,64].

An increased expression of ADAM17 in tumor-associated fibroblasts has already been described in breast cancer [65]. Therefore, it is likely that not only the tumor itself but also the surrounding microenvironment expresses ADAM17 and releases it into the bloodstream as a result of inflammation and response to the invasive growth of the carcinoma. This would explain the high ADAM17 levels in the blood, despite a low tumor burden. This requires additional studies for further investigation.

## 5. Conclusions

ADAM17 appears to be a promising screening marker for detecting early-stage ovarian cancer. The familiar markers for ovarian cancer need to also be considered. Furthermore, a combination of ADAM17 and these markers need to be evaluated. In further studies, it is conceivable to extend existing tumor markers by ADAM17 to allow more rapid early-stage detection.

## Figures and Tables

**Figure 1 cancers-13-05563-f001:**
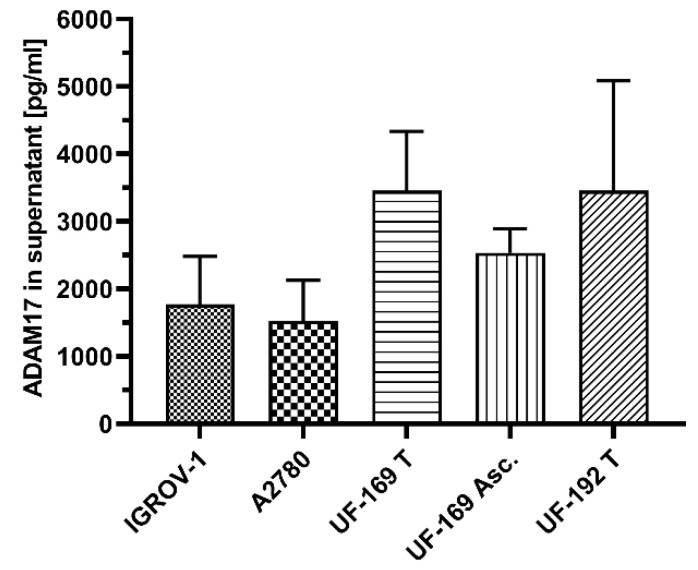
Detection of ADAM17 in CM of cultured cell lines IGROV-1, A2780 and patient-derived tumor cells isolated from tumor tissue and ascites Three independent biological experiments were carried out with two technical replicates each. No statistically significant correlation between the individual cell lines (ANOVA using Kruskal–Wallis test *p* = 0.35). Data presented as the mean + SEM. ADAM17 concentrations: IGROV-1: 1772 pg/mL, A2780: 1522 pg/mL, UF-169 T-tumor: 3456 pg/mL, UF-169 Asc: 2535 pg/mL, UF-192 T: 3457 pg/mL.

**Figure 2 cancers-13-05563-f002:**
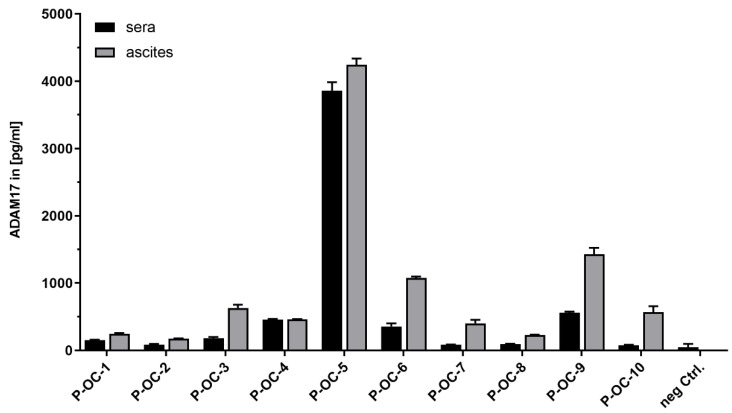
Detection of ADAM17 in ascites and in the serum of patients. Measured ADAM17 concentrations in ascites and serum were matched for each patient for better comparability. ADAM17 levels in ascites samples were on average 2.5 times higher than ADAM17 amounts in serum (*p* = 0.0049, Wilcoxon rank test). As a negative control, we used serum from a non-age-matched female donor (neg. Ctrl.).

**Figure 3 cancers-13-05563-f003:**
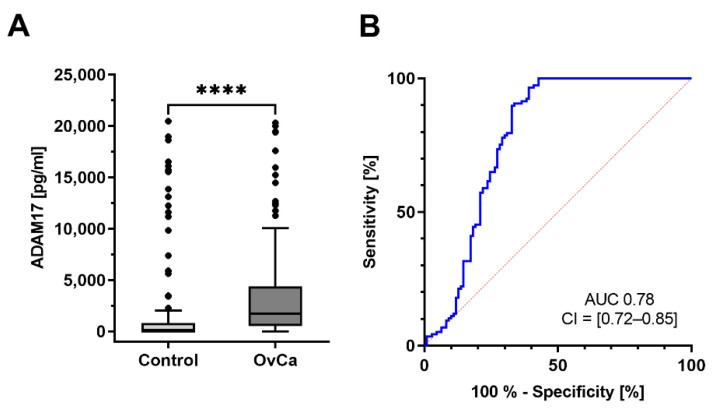
(**A**): Significant difference of ADAM17 blood levels between patient cohort and control group. One hundred blood samples from healthy donors were tested against 117 blood samples from patients diagnosed with ovarian cancer. ADAM17 levels were significantly higher in the patients’ group than in the control group. Mean average: Control: 2197 pg/mL, patients: 3753 pg/mL. **** (*p* < 0.0001). (**B**): Receiver operating characteristic (ROC): ADAM17 cut-off value of 39.792 pg/mL with a calculated sensitivity of 96.58% and specificity of 60.91% with an AUC of 0.78 (CI = [0.72–0.85]).

**Figure 4 cancers-13-05563-f004:**
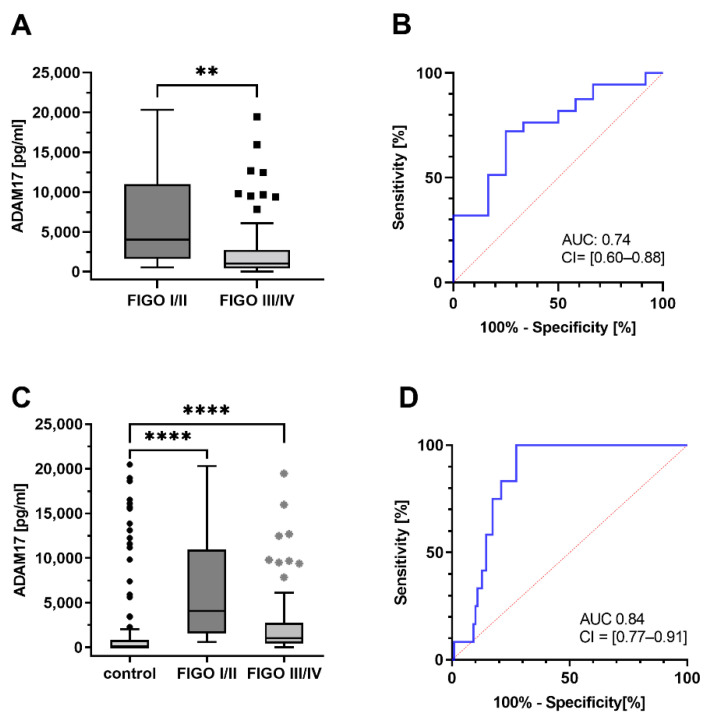
(**A**): Higher concentrations of ADAM17 in early FIGO stages. There is a significant difference between FIGO I/II and FIGO III/IV regarding the ADAM17 levels (*U*-test, *p* = 0.007) ** (*p* < 0.01). (**B**): Receiver operating characteristic: A cut-off value of 2588.5 pg/mL was calculated by ROC analysis. This corresponds to a sensitivity of 72.22% and specificity of 75.00% with an AUC of 0.74 (CI = [0.60–0.88]) predicting tumor stage in ovarian cancer at the preoperative point in time. (**C**): Comparison between healthy cohort, FIGO I/II and FIGO III/IV (*U*-test, *p* < 0.0001) **** (*p* < 0.001). (**D**): Receiver operating characteristic: AUC 0.84 (CI = [0.77–0.91]).

**Figure 5 cancers-13-05563-f005:**
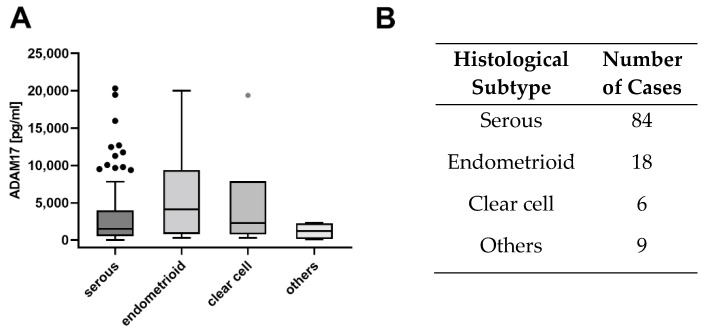
(**A**) Comparison of the individual histological subtypes. Even though there is no significant difference detectable between the individual histological subgroups, ADAM17 levels in endometrioid subtype are notedly higher compared to other subtypes. (Kruskal–Wallis test, *p* = 0.082). (**B**): Numerical description of the individual histological subtypes.

**Figure 6 cancers-13-05563-f006:**
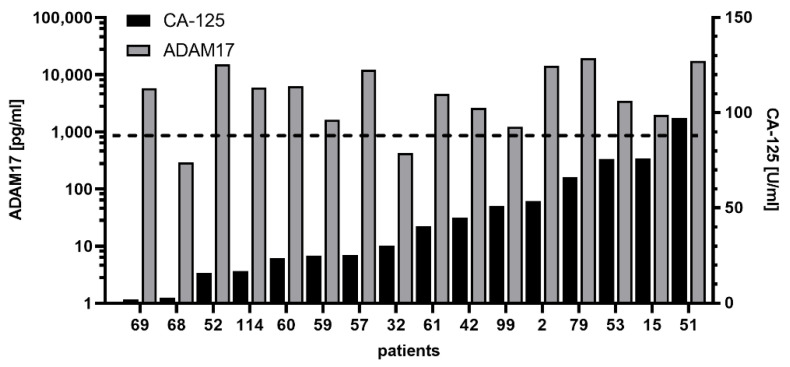
ADAM17 levels in patients with low CA-125. Each number represents a patient from the patient cohort. Descriptive bar chart showing ADAM17 level (white bars) in patients with low CA-125 (<100 units/mL, grey bars). Ordered from low to high CA-125 amounts. The dashed horizontal line indicates 75th percentile of control group.

**Table 1 cancers-13-05563-t001:** Clinical parameters of the patient cohort. Shown in the diagram: Histology of tumor, tumor grading, FIGO stage, resection of tumor mass under surgery and age.

Clinical Parameter	Characteristics			
Histology	Serous	Endometrioid	Clear-cell	Others
	84	18	6	9
Grading	Low	High		
	3	101		
FIGO	I	II	III	IV
	21	6	77	16
Resection of the tumor	R0	R1		
	79	47		
Age	Mean 60.6 +/10.8 years			

## Data Availability

The data presented in this study are available on request from the corresponding author.

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
