# Peer review of "ADAM17—A Potential Blood-Based Biomarker for Detection of Early-Stage Ovarian Cancer"

_cancers, 2021, doi:10.3390/cancers13215563_

Round 1
Reviewer 1 Report
Nina Hedemann and co-authors reported that ADAM17 could be a specific marker for early stages of ovarian cancer. They determined ADAM17 expressions in both ovarian cell lines and tumor tissue of patients. Moreover, they found that the ADAM17 levels in ascites were higher than those in serum of patients with ovarian cancer. This study contains some interesting observations. However, the manuscript needs to be major revised in order to be published.
- The title “ADAM17 - a potential biomarker for blood-based detection of early ovarian” should be improved.
- All tables (Table 1 and Figure 5B) should be the three-line.
- How to obtain the cut-off values for ADAM17 in Figure 3B and Figure 4B, what is the cut-off value for ADAM17 in Figure 4D, are they the optimal cut-point value?
- The English writing should be improved. The abbreviation (ROMA and ADAM17) should be written it out on first use; The Figure 3, 4 and 5 contain multiple sub-figures, which should be describe in the Section of Results in proper order. The word “sera” could be replaced with “serum”……
Reviewer 2 Report
In the present study, Rogmans et al. studied ADAM17 as a potential biomarker for ovarian cancer. Since ovarian cancer has the highest mortality rate among gynecological tumors, it is important to improve early detection. The authors were focused on ADAM17, based on previous reports. They concluded that ADAM17 appears to be a promising screening marker for detecting early-stage ovarian cancer. However, the experimental data does not enough support their claim.
Major problems:
1: In Figure 1, the authors did not provide information regarding the number of replicate experiments, biological replicates, and the statistical significance.
2: In Figure 2, the negative control is missing. Although the authors found that different amounts of ADAM17 are expressed in sera vs. ascites of the individual patients, this generally makes difficult to develop ADAM17 as a marker.
3: Figure 3B is not explained in the text.
4: ADAM17 protein expression is only detected by ELISA. This must be supported with multiple other methods. Is it possible to see the immunostaining of the pathological sections?
5: Related to the 4th comment, I would also see the mRNA expression level of ADAM17 and the levels of downstream substrates, such as Nectin-4 and HB-EGF.
6: Is the ADAM17 expression level associated with the prognosis?
Reviewer 3 Report
The manuscript "ADAM17 - a potential biomarker for blood-based detection of early ovarian cancer" by Christoph Rogmans et al. is a direct and well-planned study that demonstrates the interest in studying ADAM17 levels for the diagnosis of early stages of ovarian cancer disease.
It satisfactorily addresses the possibility of differentiating between healthy and sick patients in the chosen cohorts, reveals different levels in different histology types of this cancer and generates experimental evidence of the suitability of using ADAM17 levels in early detection, with advantages over the use of CA-125 in FIGO stages I-II.
The weak point for its effective clinical application is that in the selection of the cohorts, samples with the possibility of having altered levels of ADAM17 due to inflammatory processes, autoimmune diseases, etc. have been excluded. Despite this, I consider that the work deserves to be published as long as a reference to this type of limitations in clinical practice is included in the discussion.
Minor considerations to keep in mind:
Please, comment in the discussion on the value of this clinical parameter in the real population, taking into account the possibility of factors other than ovarian cancer that also progress with increased levels of ADAM17
Round 2
Reviewer 1 Report
In the new version, the authors addressed most of my comments and concerns regarding the manuscript. The revised manuscript has visibly been improved. The English writing on the section of Introduction could be improved, such as line 148-149, line 156-157, line 158-16, line 174-175, line 189-190 line 253-254. The manuscript should be accepted for publication after minor revision.
Reviewer 2 Report
I think the revised manuscript is fine. I have no further comment on this manuscript.
Author Response
We thank the reviewer for the positive feedback.